# Accelerated First-order Methods for Geodesically Convex Optimization on Riemannian Manifolds

**Yuanyuan Liu[1], Fanhua Shang[1]\*, James Cheng[1], Hong Cheng[2], Licheng Jiao[3]**
[1]Dept. of Computer Science and Engineering, The Chinese University of Hong Kong
[2]Dept. of Systems Engineering and Engineering Management,
The Chinese University of Hong Kong, Hong Kong
[3]Key Laboratory of Intelligent Perception and Image Understanding of Ministry of Education,
School of Artificial Intelligence, Xidian University, China
{yyliu, fhshang, jcheng}@cse.cuhk.edu.hk; hcheng@se.cuhk.edu.hk;
lchjiao@mail.xidian.edu.cn

## Abstract

In this paper, we propose an accelerated first-order method for geodesically convex optimization, which is the generalization of the standard Nesterov's accelerated method from Euclidean space to nonlinear Riemannian space. We first derive two equations and obtain two nonlinear operators for geodesically convex optimization instead of the linear extrapolation step in Euclidean space. In particular, we analyze the global convergence properties of our accelerated method for geodesically strongly-convex problems, which show that our method improves the convergence rate from $O((1-\mu/L)^k)$ to $O((1-\sqrt{\mu/L})^k)$. Moreover, our method also improves the global convergence rate on geodesically general convex problems from $O(1/k)$ to $O(1/k^2)$. Finally, we give a specific iterative scheme for matrix Karcher mean problems, and validate our theoretical results with experiments.

## 1 Introduction

In this paper, we study the following Riemannian optimization problem:

$$\min f(x) \quad \text{such that} \ \ x \in \mathcal{X} \subset \mathcal{M}, \tag{1}$$

where $(\mathcal{M}, \varrho)$ denotes a Riemannian manifold with the Riemannian metric $\varrho$, $\mathcal{X} \subset \mathcal{M}$ is a nonempty, compact, geodesically convex set, and $f : \mathcal{X} \to \mathbb{R}$ is *geodesically* convex ($G$-convex) and *geodesically* $L$-smooth ($G$-$L$-smooth). Here, $G$-convex functions may be non-convex in the usual Euclidean space but convex along the manifold, and thus can be solved by a global optimization solver. [5] presented $G$-convexity and $G$-convex optimization on geodesic metric spaces, though without any attention to global complexity analysis. As discussed in [11], the topic of "geometric programming" may be viewed as a special case of $G$-convex optimization. [25] developed theoretical tools to recognize and generate $G$-convex functions as well as cone theoretic fixed point optimization algorithms. However, none of these three works provided a global convergence rate analysis for their algorithms. Very recently, [31] provided the global complexity analysis of first-order algorithms for $G$-convex optimization, and designed the following *Riemannian* gradient descent rule:

$$x_{k+1} = \text{Exp}_{x_k}(-\eta \, \text{grad} f(x_k)),$$

where $k$ is the iterate index, $\text{Exp}_{x_k}$ is an exponential map at $x_k$ (see Section 2 for details), $\eta$ is a step-size or learning rate, and $\text{grad} f(x_k)$ is the Riemannian gradient of $f$ at $x_k \in \mathcal{X}$.

---

In this paper, we extend the Nesterov's accelerated gradient descent method [19] from *Euclidean* space to nonlinear *Riemannian* space. Below, we first introduce the Nesterov's method and its variants for convex optimization on Euclidean space, which can be viewed as a special case of our method, when $\mathcal{M} = \mathbb{R}^d$, and $\varrho$ is the Euclidean inner product. Nowadays many real-world applications involve large data sets. As data sets and problems are getting larger in size, accelerating first-order methods is of both practical and theoretical interests. The earliest first-order method for minimizing a convex function $f$ is perhaps the gradient method. Thirty years ago, Nesterov [19] proposed an accelerated gradient method, which takes the following form: starting with $x_0$ and $y_0 = x_0$, and for any $k \geq 1$,

$$
\begin{aligned}
x_k &= y_{k-1} - \eta \nabla f(y_{k-1}), \\
y_k &= x_k + \tau_k(x_k - x_{k-1}),
\end{aligned}
\tag{2}
$$

where $0 \leq \tau_k \leq 1$ is the momentum parameter. For a fixed step-size $\eta = 1/L$, where $L$ is the Lipschitz constant of $\nabla f$, this scheme with $\tau_k = (k-1)/(k+2)$ exhibits the optimal convergence rate, $f(x_k) - f(x_\star) \leq O(\frac{L\|x_\star - x_0\|^2}{k^2})$, for general convex (or non-strongly convex) problems [20], where $x_\star$ is any minimizer of $f$. In contrast, standard gradient descent methods can only achieve a convergence rate of $O(1/k)$. We can see that this improvement relies on the introduction of the momentum term $\tau_k(x_k - x_{k-1})$ as well as the particularly tuned coefficient $(k-1)/(k+2) \approx 1 - 3/k$. Inspired by the success of the Nesterov's momentum, there has been much work on the development of first-order accelerated methods, see [2, 8, 21, 26, 27] for example. In addition, for strongly convex problems and setting $\tau_k \equiv (1 - \sqrt{\mu/L})/(1 + \sqrt{\mu/L})$, Nesterov's accelerated gradient method attains a convergence rate of $O((1 - \sqrt{\mu/L})^k)$, while standard gradient descent methods achieve a linear convergence rate of $O((1 - \mu/L)^k)$. It is then natural to ask whether our accelerated method in nonlinear Riemannian space has the same convergence rates as its Euclidean space counterparts (e.g., Nesterov's accelerated method [20])?

## 1.1 Motivation and Challenges

Zhang and Sra [31] proposed an efficient Riemannian gradient descent (RGD) method, which attains the convergence rates of $O((1 - \mu/L)^k)$ and $O(1/k)$ for geodesically strongly-convex and geodesically convex problems, respectively. Hence, there still remain gaps in convergence rates between RGD and the Nesterov's accelerated method.

As discussed in [31], a long-time question is whether the famous Nesterov's accelerated gradient descent algorithm has a counterpart in nonlinear Riemannian spaces. Compared with standard gradient descent methods in Euclidean space, Nesterov's accelerated gradient method involves a linear extrapolation step: $y_k = x_k + \tau_k(x_k - x_{k-1})$, which can improve its convergence rates for both strongly convex and non-strongly convex problems. It is clear that $\varphi_k(x) := f(y_k) + \langle \nabla f(y_k), x - y_k \rangle$ is a linear function in Euclidean space, while its counterpart in nonlinear space, e.g., $\varphi_k(x) := f(y_k) + \langle \mathrm{grad} f(y_k), \mathrm{Exp}_{y_k}^{-1}(x) \rangle_{y_k}$, is a nonlinear function, where $\mathrm{Exp}_{y_k}^{-1}$ is the inverse of the exponential map $\mathrm{Exp}_{y_k}$, and $\langle \cdot, \cdot \rangle_y$ is the inner product (see Section 2 for details). Therefore, in nonlinear Riemannian spaces, there is no trivial analogy of such a linear extrapolation step. In other words, although Riemannian geometry provides tools that enable generalization of Euclidean algorithms mentioned above [1], we must overcome some fundamental geometric hurdles to analyze the global convergence properties of our accelerated method as in [31].

## 1.2 Contributions

To answer the above-mentioned open problem in [31], in this paper we propose a general accelerated first-order method for nonlinear *Riemannian* spaces, which is in essence the generalization of the standard Nesterov's accelerated method. We summarize the key contributions of this paper as follows.

- We first present a general Nesterov's accelerated iterative scheme in nonlinear Riemannian spaces, where the linear extrapolation step in (2) is replaced by a nonlinear operator. Furthermore, we derive two equations and obtain two corresponding nonlinear operators for both geodesically strongly-convex and geodesically convex cases, respectively.
- We provide the global convergence analysis of our accelerated algorithms, which shows that our algorithms attain the convergence rates of $O((1 - \sqrt{\mu/L})^k)$ and $O(1/k^2)$ for geodesically strongly-convex and geodesically convex objectives, respectively.

- Finally, we present a specific iterative scheme for matrix Karcher mean problems. Our experimental results verify the effectiveness and efficiency of our accelerated method.

## 2 Notation and Preliminaries

We first introduce some key notations and definitions about Riemannian geometry (see [23, 30] for details). A Riemannian manifold $(\mathcal{M}, \varrho)$ is a real smooth manifold $\mathcal{M}$ equipped with a Riemannian metric $\varrho$. Let $\langle w_1, w_2 \rangle_x = \varrho_x(w_1, w_2)$ denote the inner product of $w_1, w_2 \in T_x\mathcal{M}$; and the norm of $w \in T_x\mathcal{M}$ is defined as $\|w\|_x = \sqrt{\varrho_x(w, w)}$, where the metric $\varrho$ induces an inner product structure in each tangent space $T_x\mathcal{M}$ associated with every $x \in \mathcal{M}$. A geodesic is a constant speed curve $\gamma : [0, 1] \to \mathcal{M}$ that is locally distance minimizing. Let $y \in \mathcal{M}$ and $w \in T_x\mathcal{M}$, then an exponential map $y = \mathrm{Exp}_x(w) : T_x\mathcal{M} \to \mathcal{M}$ maps $w$ to $y$ on $\mathcal{M}$, such that there is a geodesic $\gamma$ with $\gamma(0) = x$, $\gamma(1) = y$ and $\dot{\gamma}(0) = w$. If there is a unique geodesic between any two points in $\mathcal{X} \subset \mathcal{M}$, the exponential map has inverse $\mathrm{Exp}_x^{-1} : \mathcal{X} \to T_x\mathcal{M}$, i.e., $w = \mathrm{Exp}_x^{-1}(y)$, and the geodesic is the unique shortest path with $\|\mathrm{Exp}_x^{-1}(y)\|_x = \|\mathrm{Exp}_y^{-1}(x)\|_y = d(x, y)$, where $d(x, y)$ is the geodesic distance between $x, y \in \mathcal{X}$. Parallel transport $\Gamma_x^y : T_x\mathcal{M} \to T_y\mathcal{M}$ maps a vector $w \in T_x\mathcal{M}$ to $\Gamma_x^y w \in T_y\mathcal{M}$, and preserves inner products and norm, that is, $\langle w_1, w_2 \rangle_x = \langle \Gamma_x^y w_1, \Gamma_x^y w_2 \rangle_y$ and $\|w_1\|_x = \|\Gamma_x^y w_1\|_y$, where $w_1, w_2 \in T_x\mathcal{M}$.

For any $x, y \in \mathcal{X}$ and any geodesic $\gamma$ with $\gamma(0) = x$, $\gamma(1) = y$ and $\gamma(t) \in \mathcal{X}$ for $t \in [0, 1]$ such that $f(\gamma(t)) \leq (1 - t)f(x) + tf(y)$, then $f$ is *geodesically* convex (*G*-convex), and an equivalent definition is formulated as follows:

$$f(y) \geq f(x) + \langle \mathrm{grad} f(x), \ \mathrm{Exp}_x^{-1}(y) \rangle_x,$$

where $\mathrm{grad} f(x)$ is the Riemannian gradient of $f$ at $x$. A function $f : \mathcal{X} \to \mathbb{R}$ is called *geodesically* $\mu$-strongly convex ($\mu$-strongly *G*-convex) if for any $x, y \in \mathcal{X}$, the following inequality holds

$$f(y) \geq f(x) + \langle \mathrm{grad} f(x), \ \mathrm{Exp}_x^{-1}(y) \rangle_x + \frac{\mu}{2}\|\mathrm{Exp}_x^{-1}(y)\|_x^2.$$

A differential function $f$ is *geodesically L-smooth* (*G-L*-smooth) if its gradient is *L*-Lipschitz, i.e.,

$$f(y) \leq f(x) + \langle \mathrm{grad} f(x), \ \mathrm{Exp}_x^{-1}(y) \rangle_x + \frac{L}{2}\|\mathrm{Exp}_x^{-1}(y)\|_x^2.$$

## 3 An Accelerated Method for Geodesically Convex Optimization

In this section, we propose a general acceleration method for geodesically convex optimization, which can be viewed as a generalization of the famous Nesterov's accelerated method from Euclidean space to Riemannian space. Nesterov's accelerated method involves a linear extrapolation step as in (2), while in nonlinear Riemannian spaces, we do not have a simple way to find an analogy to such a linear extrapolation. Therefore, some standard analysis techniques do not work in nonlinear space. Motivated by this, we derive two equations to bridge the gap for both geodesically strongly-convex and geodesically convex cases, and then generalized Nesterov's algorithms are proposed for geodesically convex optimization by solving these two equations.

We first propose to replace the classical Nesterov's scheme in (2) with the following update rules for geodesically convex optimization in *Riemannian* space:

$$
\begin{aligned}
x_k &= \mathrm{Exp}_{y_{k-1}}(-\eta\, \mathrm{grad} f(y_{k-1})), \\
y_k &= \mathbb{S}(y_{k-1}, x_k, x_{k-1}),
\end{aligned}
\tag{3}
$$

where $y_k, x_k \in \mathcal{X}$, $\mathbb{S}$ denotes a nonlinear operator, and $y_k = \mathbb{S}(y_{k-1}, x_k, x_{k-1})$ can be obtained by solving the two proposed equations (see (4) and (5) below, which can be used to deduce the key analysis tools for our convergence analysis) for strongly *G*-convex and general *G*-convex cases, respectively. Different from the Riemannian gradient descent rule (e.g., $x_{k+1} = \mathrm{Exp}_{x_k}(-\eta\, \mathrm{grad} f(x_k))$), the Nesterov's accelerated technique is introduced into our update rule of $y_k$. Compared with the Nesterov's scheme in (2), the main difference is the update rule of $y_k$. That is, our update rule for $y_k$ is an implicit iteration process as shown below, while that of (2) is an explicit iteration one.

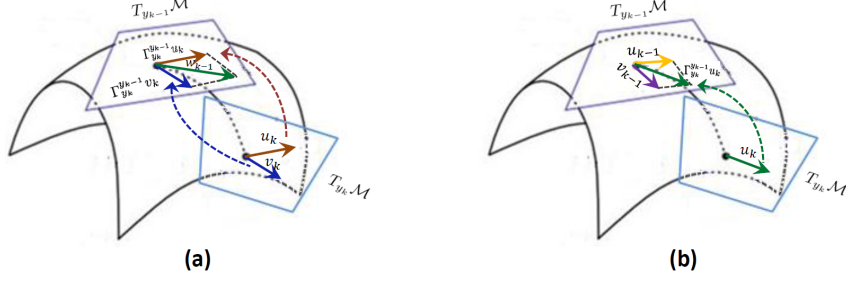

Figure 1: Illustration of geometric interpretation for Equations (4) and (5).

---

**Algorithm 1** Accelerated method for strongly $G$-convex optimization
___
**Input:** $\mu, L$
**Initialize:** $x_0, y_0, \eta$.
 1: **for** $k = 1, 2, \ldots, K$ **do**
 2:     Computing the gradient at $y_{k-1}$: $g_{k-1} = \mathrm{grad} f(y_{k-1})$;
 3:     $x_k = \mathrm{Exp}_{y_{k-1}}(-\eta g_{k-1})$;
 4:     $y_k = \mathbb{S}(y_{k-1}, x_k, x_{k-1})$ by solving (4).
 5: **end for**
**Output:** $x_K$

---

### 3.1 Geodesically Strongly Convex Cases

We first design the following equation with respect to $y_k \in \mathcal{X}$ for the $\mu$-strongly $G$-convex case:

$$\left(1 - \sqrt{\mu/L}\right) \Gamma^{y_{k-1}}_{y_k} \mathrm{Exp}^{-1}_{y_k}(x_k) - \beta \Gamma^{y_{k-1}}_{y_k} \mathrm{grad} f(y_k) = \left(1 - \sqrt{\mu/L}\right)^{3/2} \mathrm{Exp}^{-1}_{y_{k-1}}(x_{k-1}), \quad (4)$$

where $\beta = 4/\sqrt{\mu L} - 1/L > 0$. Figure 1(a) illustrates the geometric interpretation of the proposed equation (4) for the strongly $G$-convex case, where $u_k = (1-\sqrt{\mu/L})\mathrm{Exp}^{-1}_{y_k}(x_k)$, $v_k = -\beta \mathrm{grad} f(y_k)$, and $w_{k-1} = (1-\sqrt{\mu/L})^{3/2}\mathrm{Exp}^{-1}_{y_{k-1}}(x_{k-1})$. The vectors $u_k, v_k \in T_{y_k}\mathcal{M}$ are parallel transported to $T_{y_{k-1}}\mathcal{M}$, and the sum of their parallel translations is equal to $w_{k-1} \in T_{y_{k-1}}\mathcal{M}$, which means that the equation (4) holds. We design an accelerated first-order algorithm for solving geodesically strongly-convex problems, as shown in Algorithm 1. In real applications, the proposed equation (4) can be manipulated into simpler forms. For example, we will give a specific equation for the averaging real symmetric positive definite matrices problem below.

### 3.2 Geodesically Convex Cases

Let $f$ be $G$-convex and $G$-$L$-smooth, the diameter of $\mathcal{X}$ be bounded by $D$ (i.e., $\max_{x,y \in \mathcal{X}} d(x,y) \leq D$), the variable $y_k \in \mathcal{X}$ can be obtained by solving the following equation:

$$\Gamma^{y_{k-1}}_{y_k}\left(\frac{k}{\alpha-1}\mathrm{Exp}^{-1}_{y_k}(x_k) - D\widehat{g}_k\right) = \frac{k-1}{\alpha-1}\mathrm{Exp}^{-1}_{y_{k-1}}(x_{k-1}) - D\widehat{g}_{k-1} + \frac{(k+\alpha-2)\eta}{\alpha-1}g_{k-1}, \quad (5)$$

where $g_{k-1} = \mathrm{grad} f(y_{k-1})$, and $\widehat{g}_k = g_k/\|g_k\|_{y_k}$, and $\alpha \geq 3$ is a given constant. Figure 1(b) illustrates the geometric interpretation of the proposed equation (5) for the $G$-convex case, where $u_k = \frac{k}{\alpha-1}\mathrm{Exp}^{-1}_{y_k}(x_k) - D\widehat{g}_k$, and $v_{k-1} = \frac{(k+\alpha-2)\eta}{\alpha-1}g_{k-1}$. We also present an accelerated first-order algorithm for solving geodesically convex problems, as shown in Algorithm 2.

### 3.3 Key Lemmas

For the Nesterov's accelerated scheme in (2) with $\tau_k = \frac{k-1}{k+2}$ (for example, the general convex case) in Euclidean space, the following result in [3, 20] plays a key role in the convergence analysis of Nesterov's accelerated algorithm.

$$\frac{2}{k+2}\langle \nabla f(y_k), z_k - x_\star \rangle - \frac{\eta}{2}\|\nabla f(y_k)\|^2 = \frac{2}{\eta(k+2)^2}\left[\|z_k - x_\star\|^2 - \|z_{k+1} - x_\star\|^2\right], \quad (6)$$

---

**Algorithm 2** Accelerated method for general $G$-convex optimization

---

**Input:** $L, D, \alpha$

**Initialize:** $x_0, y_0, \eta$.

 1: **for** $k = 1, 2, \ldots, K$ **do**

 2:    Computing the gradient at $y_{k-1}$: $g_{k-1} = \mathrm{grad} f(y_{k-1})$ and $\hat{g}_{k-1} = g_{k-1}/\|g_{k-1}\|_{y_{k-1}}$;

 3:    $x_k = \mathrm{Exp}_{y_{k-1}}(-\eta g_{k-1})$;

 4:    $y_k = \mathbb{S}(y_{k-1}, x_k, x_{k-1})$ by solving (5).

 5: **end for**

**Output:** $x_K$

---

where $z_k = (k+2)y_k/2 - (k/2)x_k$. Correspondingly, we can also obtain the following analysis tools for our convergence analysis using the proposed equations (4) and (5). In other words, the following equations (7) and (8) can be viewed as the Riemannian space counterparts of (6).

**Lemma 1** (Strongly $G$-convex). *If $f : \mathcal{X} \to \mathbb{R}$ is geodesically $\mu$-strongly convex and $G$-$L$-smooth, and $\{y_k\}$ satisfies the equation (4), and $z_k$ is defined as follows:*

$$z_k = \left(1 - \sqrt{\mu/L}\right) \mathrm{Exp}_{y_k}^{-1}(x_k) \in T_{y_k}\mathcal{M}.$$

*Then the following results hold:*

$$\Gamma_{y_k}^{y_{k-1}}\left(z_k - \beta \mathrm{grad} f(y_k)\right) = \left(1 - \sqrt{\mu/L}\right)^{1/2} z_{k-1},$$

$$-\langle \mathrm{grad} f(y_k), z_k \rangle_{y_k} + \frac{\beta}{2}\|\mathrm{grad} f(y_k)\|_{y_k}^2 = \frac{1}{2\beta}\left(1 - \sqrt{\mu/L}\right)\|z_{k-1}\|_{y_{k-1}}^2 - \frac{1}{2\beta}\|z_k\|_{y_k}^2. \qquad (7)$$

For general $G$-convex objectives, we have the following result.

**Lemma 2** (General $G$-convex). *If $f : \mathcal{X} \to \mathbb{R}$ is $G$-convex and $G$-$L$-smooth, the diameter of $\mathcal{X}$ is bounded by $D$, and $\{y_k\}$ satisfies the equation (5), and $z_k$ is defined as*

$$z_k = \frac{k}{\alpha - 1}\mathrm{Exp}_{y_k}^{-1}(x_k) - D\hat{g}_k \in T_{y_k}\mathcal{M}.$$

*Then the following results hold:*

$$\Gamma_{y_{k+1}}^{y_k} z_{k+1} = z_k + \frac{(k+\alpha-1)\eta}{\alpha-1}\mathrm{grad} f(y_k),$$

$$\frac{\alpha-1}{k+\alpha-1}\langle \mathrm{grad} f(y_k), -z_k \rangle_{y_k} - \frac{\eta}{2}\|\mathrm{grad} f(y_k)\|_{y_k}^2 = \frac{2(\alpha-1)^2}{\eta(k+\alpha-1)^2}\left(\|z_k\|_{y_k}^2 - \|z_{k+1}\|_{y_{k+1}}^2\right). \qquad (8)$$

The proofs of Lemmas 1 and 2 are provided in the Supplementary Materials.

## 4   Convergence Analysis

In this section, we analyze the global convergence properties of the proposed algorithms (i.e., Algorithms 1 and 2) for both geodesically strongly convex and general convex problems.

**Lemma 3.** *If $f : \mathcal{X} \to \mathbb{R}$ is $G$-convex and $G$-$L$-smooth for any $x \in \mathcal{X}$, and $\{x_k\}$ is the sequence produced by Algorithms 1 and 2 with $\eta \le 1/L$, then the following result holds:*

$$f(x_{k+1}) \le f(x) + \langle \mathrm{grad} f(y_k), -\mathrm{Exp}_{y_k}^{-1}(x) \rangle_{y_k} - \frac{\eta}{2}\|\mathrm{grad} f(y_k)\|_{y_k}^2.$$

The proof of this lemma can be found in the Supplementary Materials. For the geodesically strongly convex case, we have the following result.

**Theorem 1** (Strongly $G$-convex). *Let $x_\star$ be the optimal solution of Problem (1), and $\{x_k\}$ be the sequence produced by Algorithm 1. If $f : \mathcal{X} \to \mathbb{R}$ is geodesically $\mu$-strongly convex and $G$-$L$-smooth, then the following result holds*

$$f(x_{k+1}) - f(x_\star) \le \left(1 - \sqrt{\mu/L}\right)^k \left[f(x_0) - f(x_\star) + \frac{1}{2\beta}\left(1 - \sqrt{\mu/L}\right)\|z_0\|_{y_0}^2\right],$$

*where $z_0$ is defined in Lemma 1.*

Table 1: Comparison of convergence rates for geodesically convex optimization algorithms.

| Algorithms | RGD [31] | RSGD [31] | Ours |
|---|---|---|---|
| Strongly $G$-convex and smooth | $O\left((1 - \min\{\frac{1}{c}, \frac{\mu}{L}\})^k\right)$ | $O(1/k)$ | $O\left((1 - \sqrt{\frac{\mu}{L}})^k\right)$ |
| General $G$-convex and smooth | $O\left(\frac{c}{c+k}\right)$ | $O\left(1/\sqrt{k}\right)$ | $O\left(1/k^2\right)$ |

The proof of Theorem 1 can be found in the Supplementary Materials. From this theorem, we can see that the proposed algorithm attains a linear convergence rate of $O((1-\sqrt{\mu/L})^k)$ for geodesically strongly convex problems, which is the same as that of its Euclidean space counterparts and significantly faster than that of non-accelerated algorithms such as [31] (i.e., $O((1-\mu/L)^k)$), as shown in Table 1. For the geodesically non-strongly convex case, we have the following result.

**Theorem 2** (General $G$-convex). *Let $\{x_k\}$ be the sequence produced by Algorithm 2. If $f: \mathcal{X} \to \mathbb{R}$ is $G$-convex and $G$-$L$-smooth, and the diameter of $\mathcal{X}$ is bounded by $D$, then*

$$f(x_{k+1}) - f(x_\star) \leq \frac{(\alpha - 1)^2}{2\eta(k + \alpha - 2)^2}\|z_0\|_{y_0}^2,$$

*where $z_0 = -D\widehat{g}_0$, as defined in Lemma 2.*

The proof of Theorem 2 can be found in the Supplementary Materials. Theorem 2 shows that for general $G$-convex objectives, our acceleration method improves the theoretical convergence rate from $O(1/k)$ (e.g., RGD [31]) to $O(1/k^2)$, which matches the optimal rate for general convex settings in Euclidean space. Please see the detail in Table 1, where the parameter $c$ is defined in [31].

## 5 Application for Matrix Karcher Mean Problems

In this section, we give a specific accelerated scheme for a type of conic geometric optimization problems [25], e.g., the matrix Karcher mean problem. Specifically, the loss function of the Karcher mean problem for a set of $N$ symmetric positive definite (SPD) matrices $\{W_i\}_{i=1}^N$ is defined as

$$f(X) := \frac{1}{2N}\sum_{i=1}^N \|\log(X^{-1/2}W_i X^{-1/2})\|_F^2, \tag{9}$$

where $X \in \mathcal{P} := \{Z \in \mathbb{R}^{d \times d}, \text{ s.t., } Z = Z^T \succ 0\}$. The loss function $f$ is known to be non-convex in Euclidean space but geodesically $2N$-strongly convex. The inner product of two tangent vectors at point $X$ on the manifold is given by

$$\langle \zeta,\ \xi \rangle_X = \text{tr}(\zeta X^{-1}\xi X^{-1}),\ \ \zeta, \xi \in T_X\mathcal{P}, \tag{10}$$

where $\text{tr}(\cdot)$ is the trace of a real square matrix. For any matrices $X, Y \in \mathcal{P}$, the Riemannian distance is defined as follows:

$$d(X, Y) = \|\log(X^{-\frac{1}{2}}YX^{-\frac{1}{2}})\|_F.$$

### 5.1 Computation of $Y_k$

For the accelerated update rules in (3) for Algorithm 1, we need to compute $Y_k$ via solving the equation (4). However, for the specific problem in (9) with the inner product in (10), we can derive a simpler form to solve $Y_k$ below. We first give the following properties:

**Property 1.** *For the loss function $f$ in* (9) *with the inner product in* (10)*, we have*

1. $\text{Exp}_{Y_k}^{-1}(X_k) = Y_k^{1/2}\log(Y_k^{-1/2}X_kY_k^{-1/2})Y_k^{1/2}$;

2. $\text{grad}f(Y_k) = \frac{1}{N}\sum_{i=1}^N Y_k^{1/2}\log(Y_k^{1/2}W_i^{-1}Y_k^{1/2})Y_k^{1/2}$;

3. $\left\langle \text{grad}f(Y_k), \text{Exp}_{Y_k}^{-1}(X_k)\right\rangle_{Y_k} = \langle U,\ V\rangle$;

4. $\|\text{grad}f(Y_k)\|_{Y_k}^2 = \|U\|_F^2$,

*where $U = \frac{1}{N}\sum_{i=1}^{N}\log(Y_k^{1/2}W_i^{-1}Y_k^{1/2}) \in \mathbb{R}^{d\times d}$, and $V = \log(Y_k^{-1/2}X_kY_k^{-1/2}) \in \mathbb{R}^{d\times d}$.*

*Proof.* In this part, we only provide the proof of Result 1 in Property 1, and the proofs of the other results are provided in the Supplementary Materials. The inner product in (10) on the Riemannian manifold leads to the following exponential map:

$$\mathrm{Exp}_X(\xi_X) = X^{\frac{1}{2}}\exp(X^{-\frac{1}{2}}\xi_X X^{-\frac{1}{2}})X^{\frac{1}{2}}, \tag{11}$$

where $\xi_X \in T_X\mathcal{P}$ denotes the tangent vector with the geometry, and tangent vectors $\xi_X$ are expressed as follows (see [17] for details):

$$\xi_X = X^{\frac{1}{2}}\mathrm{sym}(\Delta)X^{\frac{1}{2}}, \ \Delta \in \mathbb{R}^{d\times d},$$

where $\mathrm{sym}(\cdot)$ extracts the symmetric part of its argument, that is, $\mathrm{sym}(A) = (A^T + A)/2$. Then we can set $\mathrm{Exp}_{Y_k}^{-1}(X_k) = Y_k^{1/2}\mathrm{sym}(\Delta_{X_k})Y_k^{1/2} \in T_{Y_k}\mathcal{P}$. By the definition of $\mathrm{Exp}_{Y_k}^{-1}(X_k)$, we have $\mathrm{Exp}_{Y_k}(\mathrm{Exp}_{Y_k}^{-1}(X_k)) = X_k$, that is,

$$\mathrm{Exp}_{Y_k}(Y_k^{1/2}\mathrm{sym}(\Delta_{X_k})Y_k^{1/2}) = X_k. \tag{12}$$

Using (11) and (12), we have

$$\mathrm{sym}(\Delta_{X_k}) = \log(Y_k^{-1/2}X_kY_k^{-1/2}) \in \mathbb{R}^{d\times d}.$$

Therefore, we have

$$\mathrm{Exp}_{Y_k}^{-1}(X_k) = Y_k^{1/2}\mathrm{sym}(\Delta_{X_k})Y_k^{1/2} = Y_k^{1/2}\log(Y_k^{-1/2}X_kY_k^{-1/2})Y_k^{1/2} = -Y_k\log(X_k^{-1}Y_k),$$

where the last equality holds due to the fact that $\log(X^{-1}YX) = X^{-1}\log(Y)X$. $\qquad\square$

Result 3 in Property 1 shows that the inner product of two tangent vectors at $Y_k$ is equal to the Euclidean inner-product of two vectors $U, V \in \mathbb{R}^{d\times d}$. Thus, we can reformulate (4) as follows:

$$\left(1-\sqrt{\frac{\mu}{L}}\right)\log(Y_k^{-\frac{1}{2}}X_kY_k^{-\frac{1}{2}}) - \frac{\beta}{N}\sum_{i=1}^{N}\log(Y_k^{\frac{1}{2}}W_i^{-1}Y_k^{\frac{1}{2}}) = \left(1-\sqrt{\frac{\mu}{L}}\right)^{\frac{3}{2}}\log(Y_{k-1}^{-\frac{1}{2}}X_{k-1}^{-1}Y_{k-1}^{-\frac{1}{2}}), \tag{13}$$

where $\beta = 4/\sqrt{\mu L} - 1/L$. Then $Y_k$ can be obtained by solving (13). From a numerical perspective, $\log(Y_k^{\frac{1}{2}}W_i^{-1}Y_k^{\frac{1}{2}})$ can be approximated by $\log(Y_{k-1}^{\frac{1}{2}}W_i^{-1}Y_{k-1}^{\frac{1}{2}})$, and then $Y_k$ is given by

$$Y_k = X_k^{\frac{1}{2}}\exp^{-1}\left[\left(1-\sqrt{\frac{\mu}{L}}\right)^{\frac{1}{2}}\log(Y_{k-1}^{-\frac{1}{2}}X_{k-1}Y_{k-1}^{-\frac{1}{2}}) + \frac{\delta\beta}{N}\sum_{i=1}^{N}\log(Y_{k-1}^{\frac{1}{2}}W_i^{-1}Y_{k-1}^{\frac{1}{2}})\right]X_k^{\frac{1}{2}}, \tag{14}$$

where $\delta = 1/(1-\sqrt{\mu/L})$, and $Y_k \in \mathcal{P}$.

## 6 Experiments

In this section, we validate the performance of our accelerated method for averaging SPD matrices under the Riemannian metric, e.g., the matrix Karcher mean problem (9), and also compare our method against the state-of-the-art methods: Riemannian gradient descent (RGD) [31] and limited-memory Riemannian BFGS (LRBFGS) [29]. The matrix Karcher mean problem has been widely applied to many real-world applications such as elasticity [18], radar signal and image processing [6, 15, 22], and medical imaging [9, 7, 13]. In fact, this problem is geodesically strongly convex, but non-convex in Euclidean space.

Other methods for solving this problem include the relaxed Richardson iteration algorithm [10], the approximated joint diagonalization algorithm [12], and Riemannian stochastic gradient descent (RSGD) [31]. Since all the three methods achieve similar performance to RGD, especially in data science applications where $N$ is large and relatively small optimization error is not required [31], we only report the experimental results of RGD. The step-size $\eta$ of both RGD and LRBFGS is selected with a line search method as in [29] (see [29] for details), while $\eta$ of our accelerated method is set to $1/L$. For the algorithms, we initialize $X$ using the arithmetic mean of the data set as in [29].

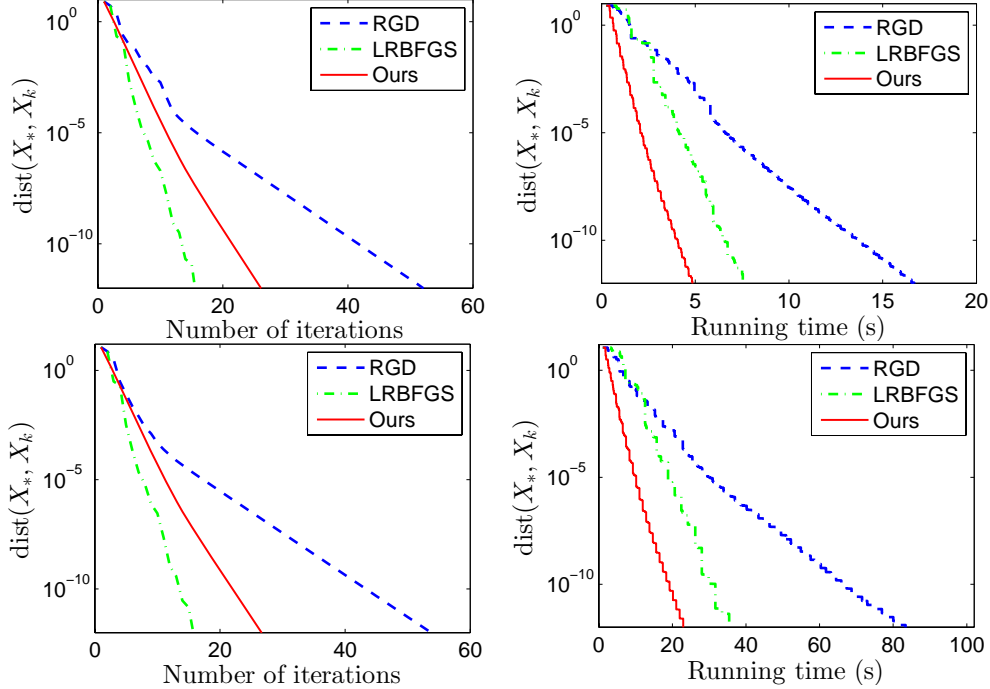

Figure 2: Comparison of RGD, LRBFGS and our accelerated method for solving geodesically strongly convex Karcher mean problems on data sets with $d = 100$ (the first row) and $d = 200$ (the second row). The vertical axis represents the distance in log scale, and the horizontal axis denotes the number of iterations (left) or running time (right).

The input synthetic data are random SPD matrices of size $100 \times 100$ or $200 \times 200$ generated by using the technique in [29] or the matrix mean toolbox [10], and all matrices are explicitly normalized so that their norms are all equal to 1. We report the experimental results of RGD, LRBFGS and our accelerated method on the two data sets in Figure 2, where $N$ is set to 100, and the condition number $C$ of each matrix $\{W_i\}_{i=1}^N$ is set to $10^2$. Figure 2 shows the evolution of the distance between the exact Karcher mean and current iterate (i.e., $\text{dist}(X_*, X_k)$) of the methods with respect to number of iterations and running time (seconds), where $X_*$ is the exact Karcher mean. We can observe that our method consistently converges much faster than RGD, which empirically verifies our theoretical result in Theorem 1 that our accelerated method has a much faster convergence rate than RGD. Although LRBFGS outperforms our method in terms of number of iterations, our accelerated method converges much faster than LRBFGS in terms of running time.

## 7   Conclusions

In this paper, we proposed a general Nesterov's accelerated gradient method for nonlinear *Riemannian* space, which is a generalization of the famous Nesterov's accelerated method for *Euclidean* space. We derived two equations and presented two accelerated algorithms for *geodesically* strongly-convex and general convex optimization problems, respectively. In particular, our theoretical results show that our accelerated method attains the same convergence rates as the standard Nesterov's accelerated method in Euclidean space for both strongly $G$-convex and $G$-convex cases. Finally, we presented a special iteration scheme for solving matrix Karcher mean problems, which in essence is non-convex in Euclidean space, and the numerical results verify the efficiency of our accelerated method.

We can extend our accelerated method to the stochastic setting using variance reduction techniques [14, 16, 24, 28], and apply our method to solve more geodesically convex problems in the future, e.g., the general $G$-convex problem with a non-smooth regularization term as in [4]. In addition, we can replace exponential mapping by computationally cheap retractions together with corresponding theoretical guarantees [31]. An interesting direction of future work is to design accelerated schemes for non-convex optimization in Riemannian space.

**Acknowledgments**

This research is supported in part by Grants (CUHK 14206715 & 14222816) from the Hong Kong RGC, the Major Research Plan of the National Natural Science Foundation of China (Nos. 91438201 and 91438103), and the National Natural Science Foundation of China (No. 61573267).

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
