[Supplementary Material]

# Supplementary Materials for "Accelerated First-order Methods for Geodesically Convex Optimization on Riemannian Manifolds"

**Yuanyuan Liu**[1], **Fanhua Shang**[1]*, **James Cheng**[1], **Hong Cheng**[2], **Licheng Jiao**[3]
[1]Dept. of Computer Science and Engineering, The Chinese University of Hong Kong
[2]Dept. of Systems Engineering and Engineering Management,
The Chinese University of Hong Kong, Hong Kong
[3]Key Laboratory of Intelligent Perception and Image Understanding of Ministry of Education,
School of Artificial Intelligence, Xidian University, China
{yyliu, fhshang, jcheng}@cse.cuhk.edu.hk; hcheng@se.cuhk.edu.hk;
lchjiao@mail.xidian.edu.cn

In this supplementary material, we give the detailed proofs for some lemmas, theorems and properties.

## A   Proof of Property 1:

**Property 1.** *The loss function $f(\cdot)$ is defined in* $(9)$ *with the inner product in* $(10)$*, then*

1. $\mathrm{Exp}_{Y_k}^{-1}(X_k) = Y_k^{1/2} \log(Y_k^{-1/2} X_k Y_k^{-1/2}) Y_k^{1/2}$;

2. $\mathrm{grad} f(Y_k) = \frac{1}{N} \sum_{i=1}^{N} Y_k^{1/2} \log(Y_k^{1/2} W_i^{-1} Y_k^{1/2}) Y_k^{1/2}$;

3. $\left\langle \mathrm{grad} f(Y_k), \ \mathrm{Exp}_{Y_k}^{-1}(X_k) \right\rangle_{Y_k} = \langle U, \ V \rangle$;

4. $\|\mathrm{grad} f(Y_k)\|_{Y_k}^2 = \|U\|_F^2$,

*where $U = \frac{1}{N} \sum_{i=1}^{N} \log(Y_k^{1/2} W_i^{-1} Y_k^{1/2}) \in \mathbb{R}^{d \times d}$, and $V = \log(Y_k^{-1/2} X_k Y_k^{-1/2}) \in \mathbb{R}^{d \times d}$.*

In the main paper, we proved the result (1) in Property 1. In the following, we will give the proofs for the results (2)-(4) in Property 1.

*Proof.* Using the derivation in [2], we have

$$\mathrm{grad} f(Y_k) = \frac{1}{N} \sum_{i=1}^{N} Y_k^{1/2} \log(Y_k^{1/2} W_i^{-1} Y_k^{1/2}) Y_k^{1/2}.$$

---

Using the results (1) and (2) and the definition of the inner product, we have

$$\left\langle \operatorname{grad} f(Y_k), \ \operatorname{Exp}_{Y_k}^{-1}(X_k) \right\rangle_{Y_k}$$

$$= \operatorname{tr}\left( \left( \frac{1}{N} \sum_{i=1}^{N} Y_k^{1/2} \log(Y_k^{1/2} W_i^{-1} Y_k^{1/2}) Y_k^{1/2} \right) Y_k^{-1} Y_k^{1/2} \log(Y_k^{-1/2} X_k Y_k^{-1/2}) Y_k^{1/2} Y_k^{-1} \right)$$

$$= \operatorname{tr}\left( Y_k^{1/2} \left( \frac{1}{N} \sum_{i=1}^{N} \log(Y_k^{1/2} W_i^{-1} Y_k^{1/2}) \right) Y_k^{1/2} Y_k^{-1} Y_k^{1/2} \log(Y_k^{-1/2} X_k Y_k^{-1/2}) Y_k^{1/2} Y_k^{-1} \right)$$

$$= \left\langle \frac{1}{N} \sum_{i=1}^{N} \log(Y_k^{1/2} W_i^{-1} Y_k^{1/2}), \ \log(Y_k^{-1/2} X_k Y_k^{-1/2}) \right\rangle$$

$$= \langle U, \ V \rangle,$$

and

$$\|\operatorname{grad} f(Y_k)\|_{Y_k}^2 = \left\| \frac{1}{N} \sum_{i=1}^{N} \log(Y_k^{1/2} W_i^{-1} Y_k^{1/2}) \right\|_F^2 = \|U\|_F^2 .$$

This completes the proof. $\qquad\square$

# B  Proofs of Lemmas 1 and 2:

Before proving Lemma 1, we first give the following property [4].

**Property 2.** *Let $\Gamma_x^y : T_x \mathcal{M} \to T_y \mathcal{M}$ be the parallel transport, and the exponential map $\operatorname{Exp}_x$ has an inverse $\operatorname{Exp}_x^{-1} : \mathcal{X} \to T_x \mathcal{M}$. For any $x, y \in \mathcal{X}$, and $w, z \in T_x \mathcal{M}$, we have the following properties:*

$$\langle w, \ z \rangle_x = \langle \Gamma_x^y w, \ \Gamma_x^y z \rangle_y ,$$
$$\|w\|_x^2 = \|\Gamma_x^y w\|_y^2,$$
$$\|\operatorname{Exp}_y^{-1}(x)\|_y^2 = \|\operatorname{Exp}_x^{-1}(y)\|_x^2 = d^2(x, y).$$

**Lemma 1** (Strongly $G$-convex). *If $f : \mathcal{X} \to \mathbb{R}$ is geodesically $\mu$-strongly convex and $G$-$L$-smooth, and $\{y_k\}$ satisfies the equation (4), and $z_k$ is defined as*

$$z_k = \left( 1 - \sqrt{\frac{\mu}{L}} \right) \operatorname{Exp}_{y_k}^{-1}(x_k) \in T_{y_k} \mathcal{M}.$$

*Then the following results hold:*

$$\Gamma_{y_k}^{y_{k-1}} \left( z_k - \beta \operatorname{grad} f(y_k) \right) = \left( 1 - \sqrt{\frac{\mu}{L}} \right)^{\frac{1}{2}} z_{k-1},$$

*and*

$$-\langle \operatorname{grad} f(y_k), z_k \rangle_{y_k} + \frac{\beta}{2} \|\operatorname{grad} f(y_k)\|_{y_k}^2 = \frac{1}{2\beta} \left( 1 - \sqrt{\frac{\mu}{L}} \right) \|z_{k-1}\|_{y_{k-1}}^2 - \frac{1}{2\beta} \|z_k\|_{y_k}^2 .$$

*Proof.* We recall the equation (4) in Algorithm 1,

$$\Gamma_{y_k}^{y_{k-1}} \left[ \left( 1 - \sqrt{\frac{\mu}{L}} \right) \operatorname{Exp}_{y_k}^{-1}(x_k) - \beta \operatorname{grad} f(y_k) \right] = \left( 1 - \sqrt{\frac{\mu}{L}} \right)^{\frac{3}{2}} \operatorname{Exp}_{y_{k-1}}^{-1}(x_{k-1}).$$

We define $z_k$ for any $k = 0, \cdots, K-1$ as follows:

$$z_k = \left(1 - \sqrt{\frac{\mu}{L}}\right) \text{Exp}_{y_k}^{-1}(x_k) \in T_{y_k}\mathcal{M}.$$

By the above two equalities, the following result holds:

$$\Gamma_{y_k}^{y_{k-1}} z_k - \Gamma_{y_k}^{y_{k-1}} \beta \text{grad} f(y_k) = \left(1 - \sqrt{\frac{\mu}{L}}\right)^{\frac{1}{2}} z_{k-1},$$

where $z_{k-1} = \left(1 - \sqrt{\frac{\mu}{L}}\right) \text{Exp}_{y_{k-1}}^{-1}(x_{k-1}) \in T_{y_{k-1}}\mathcal{M}$. Furthermore, by the above equality and the fact that $\|u\|_x^2 = \|\Gamma_x^y u\|_y^2$, then we have

$$\|z_k - \beta \text{grad} f(y_k)\|_{y_k}^2 = \|\Gamma_{y_k}^{y_{k-1}} (z_k - \beta \text{grad} f(y_k))\|_{y_{k-1}}^2$$

$$= \left(1 - \sqrt{\frac{\mu}{L}}\right) \|z_{k-1}\|_{y_{k-1}}^2.$$

By simple algebraic manipulations, we have

$$- \langle \text{grad} f(y_k), \ z_k \rangle_{y_k} + \frac{\beta}{2} \|\text{grad} f(y_k)\|_{y_k}^2 = \frac{1}{2\beta} \left(1 - \sqrt{\frac{\mu}{L}}\right) \|z_{k-1}\|_{y_{k-1}}^2 - \frac{1}{2\beta} \|z_k\|_{y_k}^2.$$

Using the above equality and the definition of $z_k$, we have

$$\left(1 - \sqrt{\frac{\mu}{L}}\right) \langle \text{grad} f(y_k), \ -\text{Exp}_{y_k}^{-1}(x_k) \rangle_{y_k} + \frac{\beta}{2} \|\text{grad} f(y_k)\|^2$$

$$= \frac{1}{2\beta} \left(1 - \sqrt{\frac{\mu}{L}}\right) \|z_{k-1}\|_{y_{k-1}}^2 - \frac{1}{2\beta} \|z_k\|_{y_k}^2.$$

This completes the proof. □

**Lemma 2.** *If $f : \mathcal{X} \to \mathbb{R}$ is G-convex and G-L-smooth, the diameter of domain is bounded by $D$, $\{y_k\}$ satisfies the equation (5), and $z_k$ is defined as*

$$z_k = \frac{k}{\alpha - 1} \text{Exp}_{y_k}^{-1}(x_k) - D\widehat{g}_k \in T_{y_k}\mathcal{M}.$$

*Then the following results hold:*

$$\Gamma_{y_{k+1}}^{y_k} z_{k+1} = z_k + \frac{(k + \alpha - 1)\eta}{\alpha - 1} \text{grad} f(y_k),$$

*and*

$$\frac{\alpha - 1}{k + \alpha - 1} \langle \text{grad} f(y_k), \ -z_k \rangle_{y_k} - \frac{\eta}{2} \|\text{grad} f(y_k)\|_{y_k}^2 = \frac{2(\alpha - 1)^2}{\eta(k + \alpha - 1)^2} \left[ \|z_k\|_{y_k}^2 - \|z_{k+1}\|_{y_{k+1}}^2 \right].$$

*Proof.* For any $k = 0, \cdots, K-1$, $z_k$ is defined as follows:

$$z_k = \frac{k}{\alpha - 1} \text{Exp}_{y_k}^{-1}(x_k) - D\widehat{g}_k,$$

where $\widehat{g}_k = \text{grad} f(y_k)/\|\text{grad} f(y_k)\|_{y_k}$. Then $z_k \in T_{y_k}\mathcal{M}$, and by the equation (5) in Algorithm 2, we have

$$\Gamma_{y_k}^{y_{k-1}} z_k = \frac{k-1}{\alpha - 1} \text{Exp}_{y_{k-1}}^{-1}(x_{k-1}) - D\widehat{g}_{k-1} + \frac{(k + \alpha - 2)\eta}{\alpha - 1} \text{grad} f(y_{k-1})$$

$$= z_{k-1} + \frac{(k + \alpha - 2)\eta}{\alpha - 1} \text{grad} f(y_{k-1}).$$

That is,
$$\Gamma^{y_k}_{y_{k+1}} z_{k+1} = z_k + \frac{(k+\alpha-1)\eta}{\alpha-1} \mathrm{grad} f(y_k).$$

Using the above equality and Property 2, we have
$$\|z_{k+1}\|^2_{y_{k+1}}$$
$$= \left\| \Gamma^{y_k}_{y_{k+1}} z_{k+1} \right\|^2_{y_k}$$
$$= \left\| z_k + \frac{(k+\alpha-1)\eta}{\alpha-1} \mathrm{grad} f(y_k) \right\|^2_{y_k}$$
$$= \|z_k\|^2_{y_k} + \frac{2(k+\alpha-1)\eta}{\alpha-1} \langle \mathrm{grad} f(y_k),\ z_k \rangle_{y_k} + \left( \frac{(k+\alpha-1)\eta}{\alpha-1} \right)^2 \|\mathrm{grad} f(y_k)\|^2_{y_k}.$$

That is,
$$-\frac{2(k+\alpha-1)\eta}{\alpha-1} \langle \mathrm{grad} f(y_k),\ z_k \rangle_{y_k} - \left( \frac{(k+\alpha-1)\eta}{\alpha-1} \right)^2 \|\mathrm{grad} f(y_k)\|^2_{y_k}$$
$$= \|z_k\|^2_{y_k} - \|z_{k+1}\|^2_{y_{k+1}}.$$

By simple algebraic manipulations and the definition of $z_k$, we have
$$\frac{2(\alpha-1)^2}{\eta(k+\alpha-1)^2} \left[ \|z_k\|^2_{y_k} - \|z_{k+1}\|^2_{y_{k+1}} \right]$$
$$= \frac{\alpha-1}{k+\alpha-1} \langle \mathrm{grad} f(y_k),\ -z_k \rangle_{y_k} - \frac{\eta}{2} \|\mathrm{grad} f(y_k)\|^2_{y_k}$$
$$= \frac{k}{k+\alpha-1} \langle \mathrm{grad} f(y_k),\ -\mathrm{Exp}^{-1}_{y_k}(x_k) \rangle_{y_k} + \frac{\alpha-1}{k+\alpha-1} \langle \mathrm{grad} f(y_k),\ D\widehat{g}_k \rangle_{y_k} - \frac{\eta}{2}\|\mathrm{grad} f(y_k)\|^2_{y_k}$$
$$= \frac{k}{k+\alpha-1} \langle \mathrm{grad} f(y_k),\ -\mathrm{Exp}^{-1}_{y_k}(x_k) \rangle_{y_k} + \frac{(\alpha-1)D}{k+\alpha-1} \|\mathrm{grad} f(y_k)\|_{y_k} - \frac{\eta}{2} \|\mathrm{grad} f(y_k)\|^2_{y_k}.$$

This completes the proof. $\qquad\square$

## C   Proof of Lemma 3:

**Lemma 3.** *Let $\{x_k\}$ be the sequence produced by Algorithms 1 and 2 with $\eta \le 1/L$. If $f : \mathcal{X} \to \mathbb{R}$ is G-convex and G-L-smooth for any $x \in \mathcal{X}$, then the following result holds:*
$$f(x_{k+1}) \le f(x) + \langle \mathrm{grad} f(y_k),\ -\mathrm{Exp}^{-1}_{y_k}(x) \rangle_{y_k} - \frac{\eta}{2} \|\mathrm{grad} f(y_k)\|^2_{y_k}.$$

*Proof.* Since $f(\cdot)$ is geodesically convex, then for any $x \in \mathcal{X}$, we have
$$f(x) \ge f(y_k) + \langle \mathrm{grad} f(y_k),\ \mathrm{Exp}^{-1}_{y_k}(x) \rangle_{y_k}. \tag{15}$$

Furthermore, $f$ is geodesically $L$-smooth, i.e.,
$$f(x) \le f(y_k) + \langle \mathrm{grad} f(y_k),\ \mathrm{Exp}^{-1}_{y_k}(x) \rangle_{y_k} + \frac{L}{2} \left\| \mathrm{Exp}^{-1}_{y_k}(x) \right\|^2_{y_k}.$$

Since $x_{k+1} = \mathrm{Exp}_{y_k}(-\eta\mathrm{grad} f(y_k)) \in \mathcal{X}$, then $\mathrm{Exp}^{-1}_{y_k}(x_{k+1}) = -\eta\mathrm{grad} f(y_k)$, and by the above equality with $x = x_{k+1}$, and the step-size $\eta \le 1/L$, we have
$$f(x_{k+1}) \le f(y_k) + \langle \mathrm{grad} f(y_k),\ \mathrm{Exp}^{-1}_{y_k}(x_{k+1}) \rangle_{y_k} + \frac{L}{2} \left\| \mathrm{Exp}^{-1}_{y_k}(x_{k+1}) \right\|^2_{y_k}$$
$$= f(y_k) - \eta\|\mathrm{grad} f(y_k)\|^2_{y_k} + \frac{L\eta^2}{2} \|\mathrm{grad} f(y_k)\|^2_{y_k} \tag{16}$$
$$\le f(y_k) - \frac{\eta}{2} \|\mathrm{grad} f(y_k)\|^2_{y_k}.$$

By (15) and (16), we have

$$f(x) - f(x_{k+1}) \geq \frac{\eta}{2} \|\text{grad} f(y_k)\|_{y_k}^2 + \left\langle \text{grad} f(y_k), \ \text{Exp}_{y_k}^{-1}(x) \right\rangle_{y_k}.$$

That is,

$$f(x_{k+1}) \leq f(x) + \left\langle \text{grad} f(y_k), \ -\text{Exp}_{y_k}^{-1}(x) \right\rangle_{y_k} - \frac{\eta}{2} \|\text{grad} f(y_k)\|_{y_k}^2, \quad \forall x \in \mathcal{M}.$$

This completes the proof. $\qquad\qquad\qquad\qquad\qquad\qquad\qquad\qquad\qquad\qquad\qquad\qquad$ $\square$

## D  Proof of Theorem 1:

**Theorem 1** (Strongly $G$-convex). *Let $x_\star$ be the optimal solution of Problem (1), and $\{x_k\}$ be the sequence produced by Algorithm 1. If $f : \mathcal{X} \to \mathbb{R}$ is geodesically $\mu$-strongly convex and $G$-$L$-smooth, then the following result holds:*

$$f(x_{k+1}) - f(x_\star) \leq \left(1 - \sqrt{\frac{\mu}{L}}\right)^k \left[f(x_0) - f(x_\star) + \frac{1}{2\beta}\left(1 - \sqrt{\frac{\mu}{L}}\right)\|z_0\|_{y_0}^2\right],$$

*where $z_0$ is defined in Lemma 1.*

*Proof.* By Lemma 3, we have

$$f(x_{k+1}) \leq f(x) + \left\langle \text{grad} f(y_k), \ -\text{Exp}_{y_k}^{-1}(x) \right\rangle_{y_k} - \frac{\eta}{2} \|\text{grad} f(y_k)\|_{y_k}^2, \quad \forall x \in \mathcal{X}. \qquad (17)$$

Let us write successively this formula at $x = x_k$ and $x = x_\star$, we have

$$f(x_{k+1}) \leq f(x_k) + \left\langle \text{grad} f(y_k), \ -\text{Exp}_{y_k}^{-1}(x_k) \right\rangle_{y_k} - \frac{\eta}{2} \|\text{grad} f(y_k)\|_{y_k}^2,$$

and

$$f(x_{k+1}) \leq f(x_\star) + \left\langle \text{grad} f(y_k), \ -\text{Exp}_{y_k}^{-1}(x_\star) \right\rangle_{y_k} - \frac{\eta}{2} \|\text{grad} f(y_k)\|_{y_k}^2. \qquad (18)$$

Furthermore, the function $f(\cdot)$ is $\mu$-strongly $G$-convex, that is,

$$f(x_\star) \geq f(y_k) + \left\langle \text{grad} f(y_k), \ \text{Exp}_{y_k}^{-1}(x_\star) \right\rangle_{y_k} + \frac{\mu}{2} \left\|\text{Exp}_{y_k}^{-1}(x_\star)\right\|_{y_k}^2.$$

Since $f(x_\star) \leq f(y_k)$, we have

$$\left\langle \text{grad} f(y_k), \ -\text{Exp}_{y_k}^{-1}(x_\star) \right\rangle_{y_k} \geq \frac{\mu}{2} \left\|\text{Exp}_{y_k}^{-1}(x_\star)\right\|_{y_k}^2.$$

Furthermore, we have

$$\frac{\mu}{2} \left\|\text{Exp}_{y_k}^{-1}(x_\star)\right\|_{y_k}^2 \leq \left\langle \text{grad} f(y_k), \ -\text{Exp}_{y_k}^{-1}(x_\star) \right\rangle_{y_k}$$
$$\leq \|\text{grad} f(y_k)\|_{y_k} \left\|\text{Exp}_{y_k}^{-1}(x_\star)\right\|_{y_k}.$$

Thus, we have

$$\frac{\mu}{2} \left\|\text{Exp}_{y_k}^{-1}(x_\star)\right\|_{y_k} \leq \|\text{grad} f(y_k)\|_{y_k}.$$

Using the above analysis, we have

$$\left\langle \text{grad} f(y_k), \ -\text{Exp}_{y_k}^{-1}(x_\star) \right\rangle_{y_k}$$
$$\leq \|\text{grad} f(y_k)\|_{y_k} \|\text{Exp}_{y_k}^{-1}(x_\star)\|_{y_k}$$
$$\leq \frac{2}{\mu} \|\text{grad} f(y_k)\|_{y_k}^2.$$

Then (18) is rewritten as follows:

$$f(x_{k+1}) \leq f(x_\star) + \left\langle \operatorname{grad} f(y_k), -\operatorname{Exp}_{y_k}^{-1}(x_\star) \right\rangle - \frac{\eta}{2} \|\operatorname{grad} f(y_k)\|^2$$

$$\leq f(x_\star) + \left(\frac{2}{\mu} - \frac{\eta}{2}\right) \|\operatorname{grad} f(y_k)\|_{y_k}^2. \tag{19}$$

Multiplying both sides of the inequality (17) by $(1 - \sqrt{\mu/L})$, and the inequality (19) by $\sqrt{\mu/L}$, then adding the two resulting inequalities with $\eta = 1/L$, we obtain

$$f(x_{k+1})$$
$$\leq \left(1 - \sqrt{\frac{\mu}{L}}\right) f(x_k) + \sqrt{\frac{\mu}{L}} f(x_\star) + \left(1 - \sqrt{\frac{\mu}{L}}\right) \left\langle \operatorname{grad} f(y_k), -\operatorname{Exp}_{y_k}^{-1}(x_k) \right\rangle_{y_k} + \frac{\beta}{2} \|\operatorname{grad} f(y_k)\|_{y_k}^2,$$

where $\beta = \frac{4}{\sqrt{\mu L}} - \frac{1}{L} > 0$. Let $z_k = \left(1 - \sqrt{\frac{\mu}{L}}\right) \operatorname{Exp}_{y_k}^{-1}(x_k)$, then by the above inequality and Lemma 1, we have

$$f(x_{k+1}) \leq \left(1 - \sqrt{\frac{\mu}{L}}\right) f(x_k) + \sqrt{\frac{\mu}{L}} f(x_\star) - \left\langle \operatorname{grad} f(y_k), z_k \right\rangle_{y_k} + \frac{\beta}{2} \|\operatorname{grad} f(y_k)\|_{y_k}^2$$

$$\leq \left(1 - \sqrt{\frac{\mu}{L}}\right) f(x_k) + \sqrt{\frac{\mu}{L}} f(x_\star) + \frac{1}{2\beta} \left(1 - \sqrt{\frac{\mu}{L}}\right) \|z_{k-1}\|_{y_{k-1}}^2 - \frac{1}{2\beta} \|z_k\|_{y_k}^2.$$

That is,

$$f(x_{k+1}) - f(x_\star)$$
$$\leq \left(1 - \sqrt{\frac{\mu}{L}}\right) [f(x_k) - f(x_\star)] + \frac{1}{2\beta} \left(1 - \sqrt{\frac{\mu}{L}}\right) \|z_{k-1}\|_{y_{k-1}}^2 - \frac{1}{2\beta} \|z_k\|_{y_k}^2. \tag{20}$$

Multiplying both sides of the inequality (20) by $(1 - \sqrt{\mu/L})^{K-k}$ and assuming $z_{-1} = z_0$, summing it over $k = 0, \cdots, K-1$, we obtain

$$f(x_K) - f(x_\star) \leq \left(1 - \sqrt{\frac{\mu}{L}}\right)^K \left[ f(x_0) - f(x_\star) + \frac{1}{2\beta} \left(1 - \sqrt{\mu/L}\right) \|z_0\|_{y_0}^2 \right].$$

This completes the proof. □

## E    Proof of Theorem 2:

**Theorem 2** (General $G$-convex)**.** *If $f : \mathcal{X} \to \mathbb{R}$ is $G$-convex and $G$-$L$-smooth, the diameter of domain is bounded by $D$, $\{x_k\}$ is produced by Algorithm 2, and let $x_\star$ be an optimal solution of Problem (1), then*

$$f(x_{k+1}) - f(x_\star) \leq \frac{(\alpha - 1)^2}{2\eta(k + \alpha - 2)^2} \|z_0\|_{y_0}^2,$$

*where $z_0$ is defined in Lemma 2.*

*Proof.* Using Lemma 3, we have

$$f(x_{k+1}) \leq f(x) + \left\langle \operatorname{grad} f(y_k), -\operatorname{Exp}_{y_k}^{-1}(x) \right\rangle_{y_k} - \frac{\eta}{2} \|\operatorname{grad} f(y_k)\|_{y_k}^2, \quad \forall x \in \mathcal{X}.$$

Let us write successively this formula at $x = x_k$ and $x = x_\star$, we obtain

$$f(x_{k+1}) \leq f(x_k) + \left\langle \operatorname{grad} f(y_k), -\operatorname{Exp}_{y_k}^{-1}(x_k) \right\rangle_{y_k} - \frac{\eta}{2} \|\operatorname{grad} f(y_k)\|_{y_k}^2, \tag{21}$$

and

$$f(x_{k+1}) \leq f(x_\star) + \left\langle \mathrm{grad}f(y_k),\ -\mathrm{Exp}_{y_k}^{-1}(x_\star) \right\rangle_{y_k} - \frac{\eta}{2}\|\mathrm{grad}f(y_k)\|_{y_k}^2,$$

$$\leq f(x_\star) + D\|\mathrm{grad}f(y_k)\|_{y_k} - \frac{\eta}{2}\|\mathrm{grad}f(y_k)\|_{y_k}^2, \tag{22}$$

where the last inequality holds due to the assumption of $d(y_k, x_\star) = \|\mathrm{Exp}_{y_k}^{-1}(x_\star)\|_{y_k} \leq D$.

Multiplying both sides of the inequality (21) by $\frac{k}{k+\alpha-1}$, and the inequality (22) by $\frac{\alpha-1}{k+\alpha-1}$, then adding the two resulting inequalities, we obtain

$$f(x_{k+1})$$
$$\leq \frac{k}{k+\alpha-1}f(x_k) + \frac{\alpha-1}{k+\alpha-1}f(x_\star) + \frac{k}{k+\alpha-1}\left\langle \mathrm{grad}f(y_k),\ -\mathrm{Exp}_{y_k}^{-1}(x_k) \right\rangle_{y_k}$$
$$+ \frac{(\alpha-1)D}{k+\alpha-1}\|\mathrm{grad}f(y_k)\|_{y_k} - \frac{\eta}{2}\|\mathrm{grad}f(y_k)\|_{y_k}^2$$
$$= \frac{k}{k+\alpha-1}f(x_k) + \frac{\alpha-1}{k+\alpha-1}f(x_\star) + \left\langle \mathrm{grad}f(y_k),\ -\frac{k}{k+\alpha-1}\mathrm{Exp}_{y_k}^{-1}(x_k) \right\rangle_{y_k}$$
$$+ \left\langle \mathrm{grad}f(y_k),\ \frac{(\alpha-1)D}{k+\alpha-1}\widehat{g}_k \right\rangle_{y_k} - \frac{\eta}{2}\|\mathrm{grad}f(y_k)\|_{y_k}^2$$
$$= \frac{k}{k+\alpha-1}f(x_k) + \frac{\alpha-1}{k+\alpha-1}f(x_\star) + \frac{\alpha-1}{k+\alpha-1}\left\langle \mathrm{grad}f(y_k),\ -z_k \right\rangle_{y_k} - \frac{\eta}{2}\|\mathrm{grad}f(y_k)\|_{y_k}^2,$$

where $\widehat{g}_k = \mathrm{grad}f(y_k)/\|\mathrm{grad}f(y_k)\|_{y_k}$. By the above inequality and Lemma 2, we have

$$f(x_{k+1}) - f(x_\star)$$
$$\leq \frac{k}{k+\alpha-1}\left[f(x_k) - f(x_\star)\right] + \frac{(\alpha-1)^2}{2\eta(k+\alpha-1)^2}\left[\|z_k\|_{y_k}^2 - \|z_{k+1}\|_{y_{k+1}}^2\right].$$

Multiplying both sides of the above inequality by $\frac{2\eta(k+\alpha-1)^2}{\alpha-1}$, we have

$$\frac{2\eta(k+\alpha-1)^2}{\alpha-1}\left[f(x_{k+1}) - f(x_\star)\right]$$
$$\leq \frac{2\eta k(k+\alpha-1)}{\alpha-1}\left[f(x_k) - f(x_\star)\right] + (\alpha-1)\left[\|z_k\|_{y_k}^2 - \|z_{k+1}\|_{y_{k+1}}^2\right].$$

It is easy to verify that $\frac{2\eta(k+\alpha-2)^2}{\alpha-1} \geq \frac{2\eta k(k+\alpha-1)}{\alpha-1}$. Summing the above inequality over $k = 0, \cdots, K-1$, we have

$$\frac{2\eta(K+\alpha-2)^2}{\alpha-1}\left[f(x_K) - f(x_\star)\right])$$
$$\leq (\alpha-1)\left[\|z_0\|_{y_0}^2 - \|z_K\|_{y_K}^2\right].$$

Multiplying both sides of the above inequality by $\frac{(\alpha-1)}{2\eta(K+\alpha-2)^2}$, we have

$$f(x_K) - f(x_\star) \leq \frac{(\alpha-1)^2}{2\eta(K+\alpha-2)^2}\|z_0\|_{y_0}^2.$$

This completes the proof. $\qquad\square$

## F  Complexity Analysis

In this part, we provide the detailed complexity analysis of our algorithm for solving the matrix Karcher mean problem. As stated in [3], the exponential mapping is expensive to compute in

practice. Therefore, we use the retraction $R_X(\xi_X) = X + \xi_X + \frac{1}{2}\xi_X X^{-1}\xi_X$ in [1] to replace with the exponential mapping in our algorithm, where $X \in \mathcal{P}$ and $\xi_X \in T_X\mathcal{P}$. Clearly, the retraction $R_X$ is cheaper to compute and tends to avoid numerical overflow [3]. Then the main cost of RGD is the computation of the Riemannian gradient, and its overall per-iteration complexity is $O((2N+3)d^3)$. Compared with RGD, our accelerated method has one more update for $Y_k$ in (14). The term $\frac{1}{N}\sum_{i=1}^{N}\log(Y_{k-1}^{1/2}W_i^{-1}Y_{k-1}^{1/2})$ can be directly available from the evaluation of the Riemannian gradient at $Y_{k-1}$, and the term $\log(Y_{k-1}^{-1/2}X_{k-1}Y_{k-1}^{-1/2})$ can also be obtained directly from (13). The cost of the computation of $Y_k$ is $O(4d^3)$, and the overall per-iteration complexity of our accelerated method is $O((2N+7)d^3)$.

For fair comparison, we implemented RGD, LRBFGS[2], and our accelerated method in Matlab, and performed all the experiments on a PC with an Intel i5-4570 CPU and 16GB RAM.

## Footnotes

[2]https://www.math.fsu.edu/~whuang2/Indices/index_Publications.html