[Reviews · NeurIPS 2017]

Reviewer 1



The paper generalizes the Nesterov’s method to geodesically convex problems on Riemannian manifolds. This is an important theoretical contribution to the field, especially it shows that such a construction is possible. Furthermore, the work on deriving the quantities for the Karcher mean problem is really interesting.

Reviewer 2



Summary of the Paper ==================== The paper considers a geodesic generalization of Nesterov's accelerated gradient descent (AGD) algorithm for Riemannian spaces. Two versions are presented: geodesic convex case and geodesic strongly convex smooth case. The proposed algorithms are instantiated for Karcher mean problems, and are shown to outperform two previous algorithms (RGD, RSGD), which address the same setting, with randomized data. Evaluation ========== From theoretical point of view, finding a proper generalization for the momentum term (so as to be able to implement AGD) which maintains the same convergence rate for any Riemannian space is novel and very interesting. From practical point of view, it is not altogether clear when the overall running time reduces. Indeed, although this algorithm requires significantly smaller number of iterations, implementing the momentum term can be very costly (as opposed to Euclidean spaces). That said, the wide range of settings to which this algorithm potentially applies makes it appealing as a general mechanism, and may encourage further development in this direction. The paper is well-written and relatively easy-to-follow. General Comments ================ - The differential geometrical notions and and other definitions used intensively throughout this paper may not be familiar for the typical NIPS reader. I would suggest making the definition section more tight and clean. In particular, the following are not seemed to be defined in the text: star-concave, star-convex, grad f(x), intrinsic inner-product, diameter of X, conic geometric optimization, retractions. - Equations 4 and 5 are given without any intuition as how should one derive them. They seem to be placed somewhat out of the blue, and I feel like the nice figure provided by the authors, which could potentially explain them, is not addressed appropriately in the text. Minor Comments ============== L52 - redundant 'The' L52+L101 - There is a great line of work which tries to give a more satisfying interpretation for AGD. Stating that the proximal interpretation as the main interpretation seems to me somewhat misleading. L59 - Can you elaborate more on the computational complexity required for implementing the exponent function and the nonlinear momentum operator S. L70 - why linearization of gradient-like updates are contributions by themselves. L72 - classes? L90 + L93 - The sentence 'we denote' is not clear. L113 - the sentence 'In addition, different..' is a bit unclear.. L126 - besides the constraint on alpha, what other considerations are needed to be taken in order to set its value optimally? L139 + L144 - Exp seems to be written in the wrong font. L151 - The wording used in Lemma 3 is a bit unclear. L155 - in Thm 1, consider restating the value of beta. L165 - How does the upper bound depend on D? Also, here again, how is alpha should be set? L177 - maybe geometrically -> geodesically? L180 - In section 5, not sure I understand how the definition of S is instantiated for this case. L181 - 'For the accelerated scheme in (4)' do you mean algorithm 1? L183 - Y_k or y_k? L193 - 'with the geometry' is not clear. L201 - Redundant 'The'. Also, (2) should be (3)? L219 - Can you explain more rigorously (or provide relevant pointers) why RGD is a good proxy for all the algorithm stated above? Page 8 - Consider providing the actual time as well, as this benchmark does not take into account the per-iteration cost. L225 - Why is C defined explicitly?

Reviewer 3



This paper gives an accelerated first-order methods for geodesically convex optimization on Riemannian manifold. It is proven tha the proposed method converge linear, i.e., O((1-\sqrt{mu/L})^k), for mu-strongly G-convex and L-smooth function and O(1/k^2) for only G-L-smooth function. This results generalize the Euclidean method: the Nesterov's accelerated method. Numerical experiments report the performance of the proposed method compared to RGD and RSGD methods. The main concern for this paper is the applications of the proposed method. The experimental result is not convincing. For Karcher mean on SPD matrices, the cost function is smooth and the Hessian always has good condition number. Methods that explore higher order information have better performance. The authors claims that Bini's method, Riemannian GD method and limited-memory Riemanian BFGS have similar performance. However, it is not the case, especially when BB step size instead of a constant step size is used as the initial step size in line search algorthm. Usually 30 passes can reduce the objective gap by a factor more than 10^10 rather than only 10^3 in this paper. This application seems not to be a good example for the proposed method. Since the propose method does not require the function to be C^2, an application with C^1 cost function may be more suitable to show the performance of the proposed method.